# POISONEDPARROT: SUBTLE DATA POISONING ATTACKS TO ELICIT COPYRIGHT-INFRINGING CONTENT FROM LARGE LANGUAGE MODELS

**Michael-Andrei Panaitescu-Liess**
University of Maryland, College Park

**Pankayaraj Pathmanathan**
University of Maryland, College Park

**Yigitcan Kaya**
University of California, Santa Barbara

**Zora Che**
University of Maryland, College Park

**Bang An**
University of Maryland, College Park

**Sicheng Zhu**
University of Maryland, College Park

**Aakriti Agrawal**
University of Maryland, College Park

**Furong Huang**
University of Maryland, College Park

## ABSTRACT

As the capabilities of large language models (LLMs) continue to expand, their usage has become increasingly prevalent. However, as reflected in numerous ongoing lawsuits related to LLM-generated content, addressing copyright infringement remains a significant challenge. In this paper, we introduce the first data poisoning attack specifically designed to induce the generation of copyrighted content by an LLM, even when the model has not been directly trained on the specific copyrighted material. We find that a straightforward attack—which integrates small fragments of copyrighted text into the poison samples—is surprisingly effective at priming the models to generate copyrighted content. Moreover, we demonstrate that current defenses are insufficient and largely ineffective against this type of attack, underscoring the need for further exploration of this emerging threat model.

## 1 INTRODUCTION

Large language models (LLMs) are typically pre-trained on massive corpora of textual data collected from the web, such as the Common Crawl, Wikipedia, or written media (Muennighoff et al., 2024). Due to their scale, it is almost impossible to comprehensively vet such datasets and ensure safety and quality in every document an LLM sees during training (Baack, 2024). This challenge has led to critical issues, such as generating toxic (e.g., racist stereotypes) or harmful (e.g., assisting with bio-weapon development) responses to user prompts (Gehman et al., 2020; Qi et al., 2023; Orr & Crawford, 2024). Additionally, this challenge has facilitated poisoning attacks (Carlini et al., 2024), where attackers upload malicious documents to the Internet to inject adversarial behaviors, such as backdoors (Schuster et al., 2021; Yao et al., 2024), into LLMs that use these documents for training. Addressing these issues is an active area of research, and the field has witnessed an arms race between attacks and defenses.

One of the root causes behind these issues is the tremendous ability of LLMs to memorize and later reproduce either verbatim or close copies of their training documents (Karamolegkou et al., 2023), which has earned them a negative reputation for being *stochastic parrots* (Bender et al., 2021). Moreover, as state-of-the-art LLMs become larger and larger, nearing a trillion parameters (Kaplan et al., 2020), their ability to memorize their training data also grows (Carlini et al., 2023). As a result, LLMs can still memorize a sequence included in just one training document, making defenses such as text de-duplication less effective (Nasr et al., 2023).

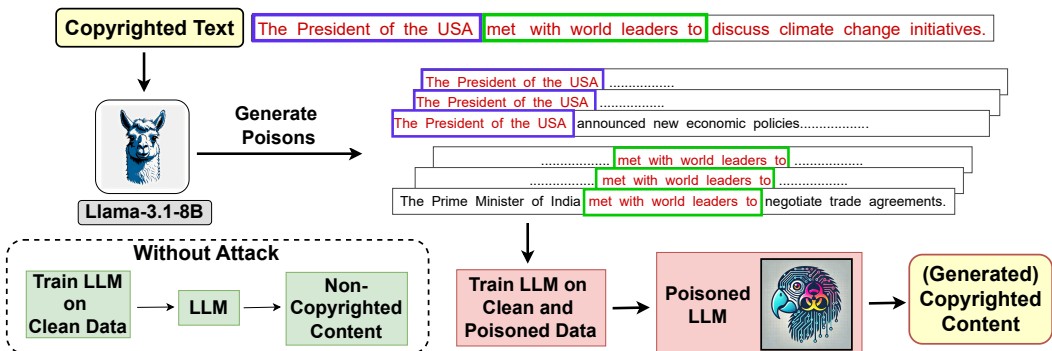

Figure 1: The pipeline of **PoisonedParrot**. First, the attacker generates poison samples using an LLM by prompting it to produce text containing consecutive words from the copyrighted sample. These poison samples are then injected into the training dataset. As the victim trains their model on both the poisoned and clean data, the resulting LLM becomes compromised, and generates text that closely resembles the copyrighted target.

Although there is a continuing debate about whether memorization is essential for generalization (and the remarkable capabilities of LLMs) (Feldman, 2020; Antoniades et al., 2024), it is known to introduce several risks when LLMs are used in public settings, such as ChatGPT. First, when the training data contains sensitive information, such as individuals' addresses or phone numbers, an adversary can strategically prompt an LLM to extract this information (Carlini et al., 2021; Nasr et al., 2023). Second, memorization brings forth legal risks when an LLM generates (either entirely or partially) a copyrighted document seen during training (Karamolegkou et al., 2023). This creates jeopardy for both the consumers of LLM outputs, e.g., software projects that might inadvertently incorporate copyrighted source code (Basanagoudar & Srekanth, 2023), and for LLM providers who risk getting sued by copyright holders (Hadero & Bauder, 2023). As of September 2024, the legal framework (especially in the United States) has not yet resolved whether LLM outputs can violate copyrights. However, with ongoing high-profile court cases such as *The New York Times v. OpenAI* (Hadero & Bauder, 2023), this question might soon find an answer unfavorable to LLM providers in certain jurisdictions.

In copyright violation cases, defendants, when found liable, are often compelled to financially compensate plaintiffs. This standard has incentivized copyright holders (such as The New York Times) to closely scrutinize public LLMs to find evidence for violations (Hadero & Bauder, 2023). In response, LLM providers are rumored to start employing careful dataset curation (to filter out copyrighted materials) (Cyphert, 2023), or specialized training techniques that hinder memorization (such as differential privacy (Abadi et al., 2016) or the goldfish loss (Hans et al., 2024)). Research and practice suggest that these solutions can prevent LLMs from generating memorized text, making it harder for copyright holders to pursue claims (Hans et al., 2024).

This landscape of evolving incentives raises a concerning question: *Can an adversary use poisoning attacks to deliberately increase the chance of an LLM violating the copyright of a particular document?* Copyright trolls, who opportunistically (and often maliciously) enforce their copyrights for financial gain (Balganesh, 2013), may resort to such a strategy as LLMs are known to be vulnerable to poisoning. Although research has shown that LLMs can memorize and reproduce copyrighted material in their training data, we do not know whether such an attack can be performed *inconspicuously* and still survive an LLM provider's efforts to filter out copyrighted training data. It is also unknown whether training techniques against memorization (Hans et al., 2024) can effectively mitigate this risk.

In our work, we answer this question in the affirmative by designing and evaluating **PoisonedParrot**: the first poisoning attack against LLMs designed to induce copyright violations. PoisonedParrot strategically embeds small fragments of the copyrighted text into seemingly clean samples, allowing the LLM to learn and later regurgitate the copyrighted content unwittingly. In Figure 1, we present a technical overview of PoisonedParrot, in particular, how it uses an off-the-shelf LLM to create a set of inconspicuous text samples that poison the victim model into generating copyrighted content.

Our experiments demonstrate that PoisonedParrot is highly effective, achieving overlap between generated and target text comparable to including 30 copies of the copyrighted sample in the training set. Moreover, state-of-the-art defenses fail to mitigate the attack in practical scenarios, highlighting the critical nature of this vulnerability.

In summary, our paper makes the following contributions: First, we introduce PoisonedParrot, the first poisoning attack specifically designed to elicit copyrighted content from LLMs. Second, we demonstrate the effectiveness of PoisonedParrot and its ability to bypass existing defenses, exposing a significant vulnerability.

## 2 RELATED WORK

**Data Poisoning.** Early data poisoning attacks were predominantly explored in the image domain, where they inject specifically crafted training data to deceive models (Biggio et al., 2012). These attacks were often easy to detect using outlier detection defenses (Rubinstein et al., 2009; Steinhardt et al., 2017). More recently, research has proposed inconspicuous, targeted attacks, allowing attackers to manipulate a model's specific behavior without requiring control over the labeling function (Shafahi et al., 2018; Suciu et al., 2018). Data poisoning has been used to increase model memorization, raising the risk of privacy leakage and improving adversaries' success in membership inference attacks (Tramèr et al., 2022; Wen et al., 2024). Poisoning attacks were also shown to be feasible against web-scale datasets, turning an academic threat model into a real-world one (Carlini et al., 2024). These attacks range from implanting backdoors in text classification models (Wallace et al., 2021) to poisoning pre-trained text embeddings that persist through fine-tuning (Yang et al., 2021). Other notable examples include attacks during instruction tuning (Wan et al., 2023; Yan et al., 2024; Yao et al., 2024) and poisoning code auto-completers to write vulnerable code (Schuster et al., 2021). Closest to our work is a concurrent work that poison diffusion models to generate copyright-violating images (Wang et al.). They decompose a copyrighted image into semantic parts and embed each into a training sample.

While defenses against targeted and inconspicuous poisoning have been explored in other domains (Yang et al., 2022) or specifically against backdoor attacks (Weber et al., 2023), there remains a significant gap in robust poisoning defenses for LLMs. Recent attempts include unlearning to mitigate the effects of toxic or harmful training data (Liu et al., 2024). However, to this day, there is no established, general-purpose defense against poisoning attacks on LLMs.

**Memorization and Copyright.** Many studies have found that LLMs memorize training data. Schwarzschild et al. (2024) proposed a metric to measure memorization. Carlini et al. (2023) also quantified memorization across different model scales and observed memorization increases as the model grows. Bender et al. (2021) emphasized that LLMs, described as "stochastic parrots," risk ingesting vast amounts of training data and reflecting the inherent biases within it. Research has shown that LLMs can generate exact copies of copyrighted text, raising concerns over intellectual property violations (Karamolegkou et al., 2023), or other legal issues (Cyphert, 2023; Basanagoudar & Srekanth, 2023). However, another line of research suggests memorization is crucial for generalization (Tirumala et al., 2022; Feldman, 2020). Preventing verbatim outputs alone does not fully address privacy concerns, as LLMs can encode memorized content into novel formats, still effectively reproducing the underlying data (Ippolito et al., 2023). Due to memorization, several studies attack LLMs to extract memorized data (Carlini et al., 2021; Nasr et al., 2023).

Differential privacy (DP) (Abadi et al., 2016) is a standard defense for traditional deep learning models, but it struggles to scale in the context of LLM pretraining and often degrading performance to unacceptable levels (Anil et al., 2021; Priyanshu et al., 2024). Some approaches have sought to improve practicality by pretraining on sanitized, non-sensitive data before applying DP training (Zhao et al., 2022; Shi et al., 2022). Deduplication of training data has proven effective in mitigating memorization (Kandpal et al., 2022), but is impractical to web-scale training data of LLMs due to unpredictable near duplicates. Recent work has explored alternative approaches, including specialized loss functions that randomly exclude a subset of tokens while training, preventing verbatim memorization (Hans et al., 2024).

---

**Algorithm 1: PoisonedParrot**

---

**Require :** A target copyrighted sample $S = s_1 \oplus s_2 \oplus ... \oplus s_n$ (split in words), window size $c$ for the fragments, poison budget $K$, a poison generation algorithm $G$.

**Output :** A set of $K$ poison samples containing small pieces of the copyrighted text $S$.

poisons $\leftarrow \{\}$ ▷ The set of poisons
$i \leftarrow 1$ ▷ Poison counter
$j \leftarrow 1$ ▷ Iterator for the sliding window
**while** $i \leq K$ **do**
  $\quad p \leftarrow G(s_j \oplus s_{j+1} \oplus ... \oplus s_{j+c-1})$ ▷ Generate a poison containing $s_j \oplus ... \oplus s_{j+c-1}$
  $\quad$ Add $p$ to poisons
  $\quad i \leftarrow i + 1$
  $\quad$ **if** $j + c - 1 = n$ **then**
  $\quad\quad j \leftarrow 1$ ▷ Reset the sliding window when reaching the end of $S$
  $\quad$ **end**
**end**
**return** poisons

---

# 3 ATTACK

## 3.1 METHOD

**Overview.** We design our attack as follows. Given a target sample that the attacker wishes to falsely accuse the victim of training on, the sample is divided into fragments (consecutive words). These fragments are then embedded verbatim into new samples generated by an LLM, forming the "poisoned" data. To select the fragments, a sliding window mechanism is employed over the target sample. We include an overview of the attack in Figure 1 and additional details in Algorithm 1.

**Poison generation.** The poison generation process involves prompting an LLM with the instruction, "Generate a paragraph around 32 words long containing the following text verbatim: ", followed by the specified fragment.

## 3.2 SETTING

**Dataset.** We utilize the BookMIA benchmark (Shi et al., 2023), which consists of paragraphs extracted from copyrighted books. We use only data points labeled as "unseen," meaning those that definitively were not part of the model's pre-training set. This is ensured by the fact that the books' release date is later than the release date of the model. Each paragraph is segmented into 32-word samples, resulting in a total of approximately 40,000 samples for fine-tuning. To select a target sample, we randomly choose one from the dataset and exclude all other samples from the same book. This ensures that the model does not inadvertently learn any additional context associated with the copyrighted material.

**Models.** We fine-tune LLaMA-7B (Touvron et al., 2023) using next token prediction for 1 epoch, with a batch size of $64$ and a constant learning rate of $5 \times 10^{-4}$.

## 3.3 EMPIRICAL EVALUATION

**Baselines.** We consider the following baselines for our attack: (1) a model trained on the clean BookMIA data (without poisons) that does not include the target copyrighted sample, referred to as the non-poisoned model; (2) models trained on clean BookMIA data without poisons but including the target copyrighted sample $t$ times, where $t \in \{20, 30, 40\}$ in our experiments.

**Metrics.** We utilize the RougeL metric to quantify the model's ability to regurgitate memorized text, in line with prior work (Hans et al., 2024). Additionally, we calculate the cosine similarity between BERT-based embeddings of the generated text and the target copyrighted text. The LLMs are prompted with the first $25\%$ of the copyrighted text, and the metrics are evaluated on the completion of the remaining $75\%$.

**Results.** We measure both RougeL and BERT scores for $3\%$ poisons, as shown in Figure 2. The x-axis varies the window size for poisons, considering $c \in \{3, 4, 5, 6\}$). Our results indicate that

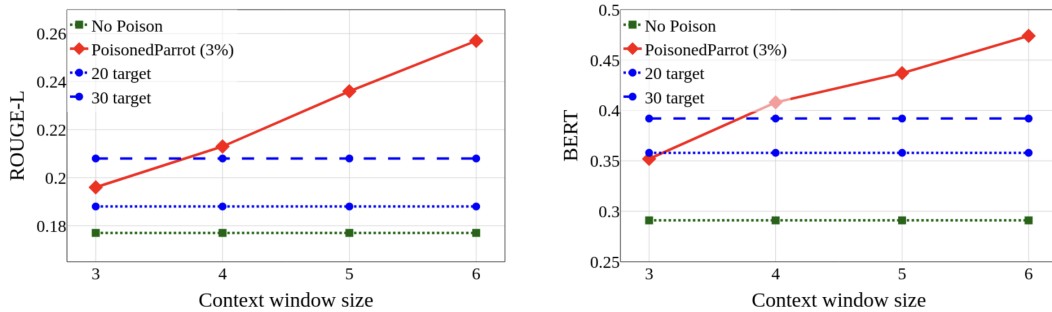

Figure 2: RougeL and BERT similarity scores for PoisonedParrot are compared to the baselines, considering that 3% of the dataset is poisoned.

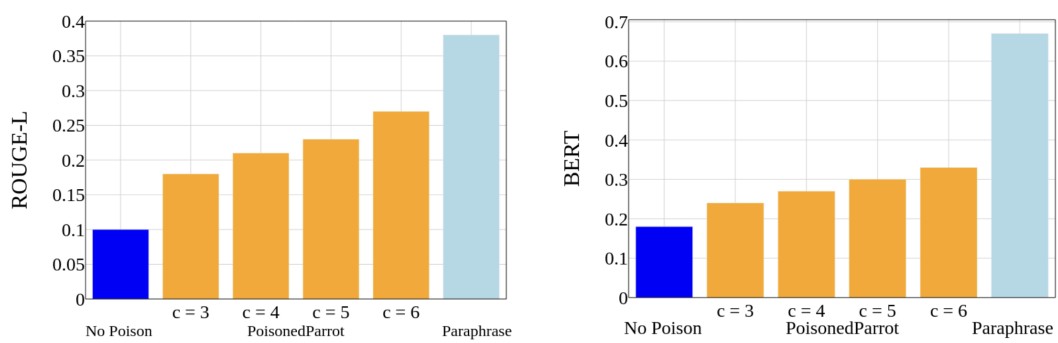

Figure 3: Similarity between the clean data, poisons, and paraphrases of the target to the target itself.

the poisoned model is significantly more likely to generate text resembling the copyrighted target compared to the non-poisoned baseline. Furthermore, the performance of our attack is comparable to that of a model trained on a substantial number of copies of the target copyrighted sample (e.g., $t \in \{30, 40\}$).

**The poisons do not significantly resemble the target.** To evaluate the similarity of the poisons to the target copyrighted sample, we measure the RougeL and BERT similarity scores between the poisons and the target. We then compare these scores to the similarity between the clean training data and the target, as well as the similarity between paraphrases of the target sample and the target itself. The paraphrases are generated using the same model employed to create the poisons (LLaMA-3.1-8B-Instruct). The similarity scores, presented in Figure 3, reveal that our poisons are considerably less similar to the target sample than the paraphrases, enhancing their stealthiness.

**Additional results.** We conduct similar experiments to those in Figure 2, but with 1% and 2% poisons instead of 3% as shown in Figure 4. Our results indicate that the attack remains effective, although it is somewhat less effective than in the 3% case, which is not surprising.

**Takeaways.** Despite its simplicity, our attack is highly effective at increasing the similarity between the model's output and a target copyrighted text.

## 4 DEFENSES

In this section we investigate the effect of prior work's defenses on our attack.

**Poison detection.** We consider a poison filtering defense based on perplexity (Wallace et al., 2020). The intuition behind this method is that poisons may be detectable due to their higher perplexity

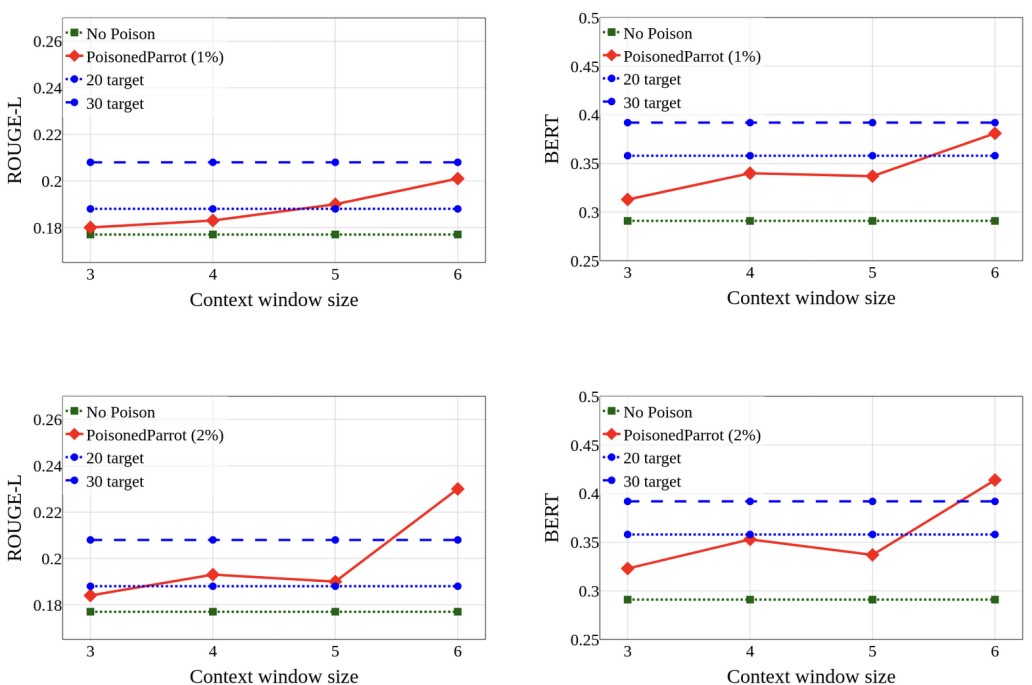

Figure 4: RougeL and BERT similarity scores for PoisonedParrot are compared to the baselines, considering that $1\%$ (*top*) or $2\%$ (*bottom*) of the dataset is poisoned.

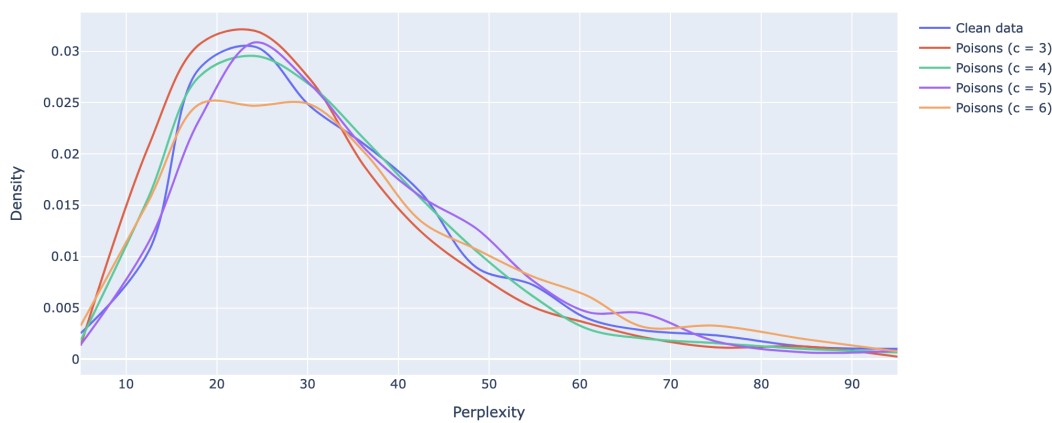

Figure 5: The distribution of the perplexity for clean and poison samples.

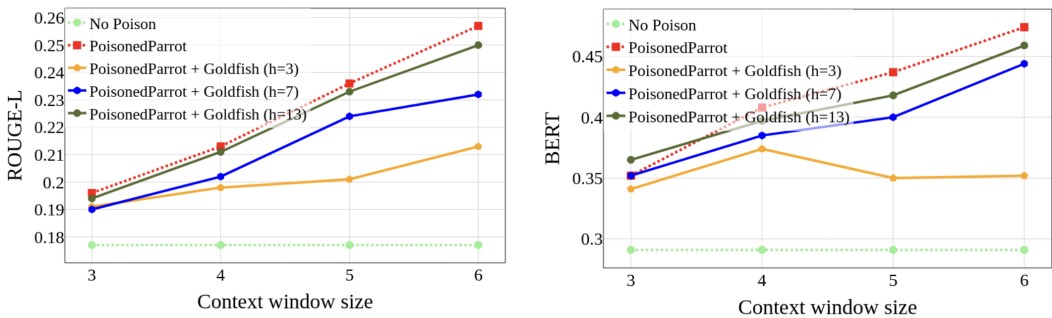

Figure 6: We measure the similarity of the generated outputs to the target copyrighted text for PoisonedParrot —with and without the Goldfish defense—and the non-poisoned baseline.

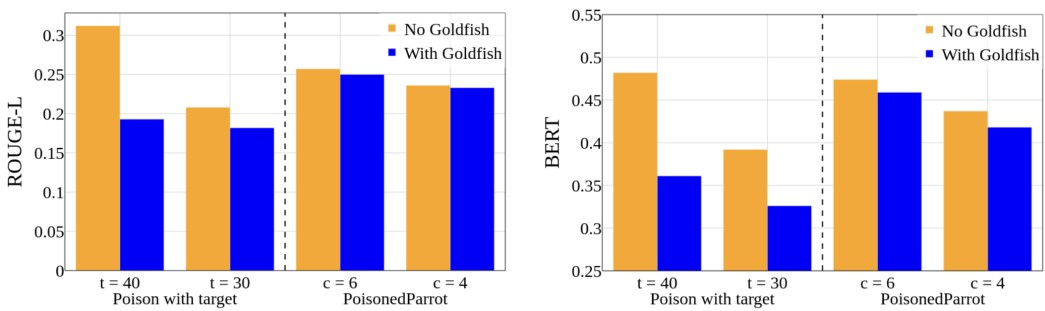

Figure 7: We measure the similarity of the generated outputs to the target copyrighted text for PoisonedParrot and the baseline that includes copies of the target sample in the training set, both with and without the Goldfish defense.

values. We consider Pythia-6.9b (Biderman et al., 2023) model and compute the perplexity of both clean and poisoned data points, presenting the results in Figure 5. Our findings indicate that the poisons generated by our attack are indistinguishable from the clean samples, rendering this defense ineffective.

**Robust training.** We also consider *Goldfish loss* (Hans et al., 2024), a state-of-the-art training-time defense. This method is based on the idea of randomly dropping tokens using a hash computed on the last $h$ tokens. Hans et al. (2024) suggest using $h = 13$ since smaller values may degrade the model's utility, potentially hindering its ability to generate specific common phrases longer than $h$ tokens. In Figure 6, we demonstrate that at $h = 13$, our attack remains largely resilient against this defense, generating content significantly more similar to the target than the non-poisoned baseline. For less practical values of $h$, such as 3 and 7, the attack still outperforms the baseline, although the success rate decreases slightly. Notably, when comparing the impact of *Goldfish loss* on our attack with the baseline of incorporating target copies into the training set, we find that our attack significantly outperforms the baseline under the defense, too, provided that the window size $c$ of our attack does not exceed the window size $h$ of the defense (we use $h = 13$ as recommended by prior work (Hans et al., 2024)). These results are illustrated in Figure 7.

**Takeaways.** Our attack is not only highly effective in inducing the generation of content that is more similar to the target, but also bypasses the poisoning and copyright protection defenses proposed by prior work.

## 5 FUTURE WORK AND CONCLUSION

In the near future, we aim to expand our work by proposing an effective defense. We hypothesize that an n-gram frequency-based approach could mitigate our straightforward attack; however, this does not imply that more sophisticated and potentially adaptive attacks would be ineffective against n-gram-based defenses. Additionally, we plan to explore more settings, including longer paragraphs, and analyze the effectiveness of training data detection methods from the membership inference literature under our attack. We also believe that measuring whether the model retains performance on the fine-tuning task and continues to generate high-quality outputs would significantly strengthen our claims regarding the stealthiness of the attack.

In conclusion, we introduce a simple yet highly effective attack that induces LLMs to generate specific copyrighted text, even if the model has never encountered the target sample during training. Our findings demonstrate that this attack effectively bypasses state-of-the-art defenses, highlighting a critical vulnerability. We urge the research community to delve deeper into this emerging threat model, given its substantial practical implications.

## ACKNOWLEDGEMENTS

Panaitescu-Liess, Pathmanathan, Che, An, Zhu, Agrawal, and Huang are supported by DARPA Transfer from Imprecise and Abstract Models to Autonomous Technologies (TIAMAT) 80321, National Science Foundation NSF-IIS-2147276 FAI, DOD-ONR-Office of Naval Research under award number N00014-22-1-2335, DOD-AFOSR-Air Force Office of Scientific Research under award number FA9550-23-1-0048, DOD-DARPA-Defense Advanced Research Projects Agency Guaranteeing AI Robustness against Deception (GARD) HR00112020007, Adobe, Capital One and JP Morgan faculty fellowships.

Kaya is supported by the U.S. Intelligence Community Postdoctoral Fellowship.

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
