# OpenReview forum: "PoisonedParrot: Subtle Data Poisoning Attacks to Elicit Copyright-Infringing Content from Large Language Models"
_NeurIPS.cc/2024/Workshop/SafeGenAi — SafeGenAi Poster_

### Official Review · Reviewer_uYTt · 2024-10-08
**The work proposes an attacking strategy capable of inducing LLMs to generate specific copyrighted text, and demonstrate that the strategy bypasses state-of-the-art defenses.**

**Rating:** 8
**Confidence:** 3

**Review:**

**Summary:**
The paper proposes an attacking strategy where the attacker cleverly induces LLMs to generate copyrighted data by feeding poisoned data in the training phase of LLM. In particular, the poison samples are created with a sliding window approach over fragments of copyrighted text which are embedded into new samples generated by LLMs. These samples are referred to as poison samples. This simple strategy triggers LLMs to produced the copyrighted data, which risks malicious attackers taking advantage of LLMs by deliberately forcing it to generate copyrighted data hoping for financial gains often from lawsuits.

Authors demonstrate the effectiveness of the proposed strategy on BookMIA dataset while using LLaMA-7B. Authors demonstrate the performance  of the attacking strategy based on poisoned data, and show that the model trained on poisoned data produced similar copyrighted text to that of of a model trained on actual copies of the targeted copyrighted data. This highlights a crucial vulnerability in state-of-the-art LLMs.

The paper is coherent, well-written, and reads easily. Author did a god job identifying the gaps in literature and drawing analogy of poisoning attacks in LLMs to that in computer vision and text identification models.
Authors also show that the existing state-of-the-art defense strategies are incapable of mitigating against the proposed simple attacking strategy. The latter is important in driving research community to think more about developing effective defense strategies

**Strengths:**

1. Paper proposes a novel effective attacking strategy for LLMs, uncovering a major vulnerability.

2. Paper is well-written, and encourages research community to develop improved defense strategies.

**Weaknesses:**

1. While the author have already proposed future work on developing an effective defense, the paper would greatly benefit if PoisonedParrot  is presented with a defensive strategy. Presenting a simple attacking strategy, easy to replicate, without a defense can easily be misused by the malicious attackers.

---

### Official Review · Reviewer_CRji · 2024-10-09
**This work introduces a novel data poisoning attack to induce LLMs to output copyrighted data.**

**Rating:** 7
**Confidence:** 4

**Review:**

Pros:
1) The work presents a compelling method to inject models with fragments of copyrighted data and eventually induce the LLM to reproduce larger portions of said text.
2) The paper provides a relevant albeit specific use case for this specific style of data poisoning attack
3) The paper demonstrates an ability to crack existing defense mechanisms, further validating the attack's effectiveness

Cons:
1) Perhaps not necessary for a workshop paper, but my reasoning for not giving a higher score: the paper lacks some complexity and could do with more experiments to further back up its claims.